# H2A histone-fold and DNA elements in nucleosome activate SWR1-mediated H2A.Z replacement in budding yeast

**Anand Ranjan[1]\*, Feng Wang[2], Gaku Mizuguchi[1], Debbie Wei[2], Yingzi Huang[2], Carl Wu[1,2]\***

[1]Janelia Research Campus, Howard Hughes Medical Institute, Ashburn, United States; [2]Laboratory of Biochemistry and Molecular Biology, National Cancer Institute, National Institutes of Health, Bethesda, United States

**Abstract** The histone variant H2A.Z is a universal mark of gene promoters, enhancers, and regulatory elements in eukaryotic chromatin. The chromatin remodeler SWR1 mediates site-specific incorporation of H2A.Z by a multi-step histone replacement reaction, evicting histone H2A-H2B from the canonical nucleosome and depositing the H2A.Z-H2B dimer. Binding of both substrates, the canonical nucleosome and the H2A.Z-H2B dimer, is essential for activation of SWR1. We found that SWR1 primarily recognizes key residues within the α2 helix in the histone-fold of nucleosomal histone H2A, a region not previously known to influence remodeler activity. Moreover, SWR1 interacts preferentially with nucleosomal DNA at superhelix location 2 on the nucleosome face distal to its linker-binding site. Our findings provide new molecular insights on recognition of the canonical nucleosome by a chromatin remodeler and have implications for ATP-driven mechanisms of histone eviction and deposition.

\*For correspondence: ranjana@janelia.hhmi.org (AR); wuc@janelia.hhmi.org (CW)

**Competing interests:** The authors declare that no competing interests exist.

## Introduction

The histone variant H2A.Z, a universal component of nucleosomes flanking eukaryotic promoters, enhancers, and other genetic elements, has an important role in transcriptional regulation (*Santisteban et al., 2000*; *Albert et al., 2007*; *Barski et al., 2007*). In *Saccharomyces cerevisiae*, H2A.Z is deposited by the ATP-dependent activity of the multi-component SWI/SNF-related SWR1 complex, which replaces nucleosomal histone H2A-H2B with H2A.Z-H2B in a coupled histone-dimer transfer (*Mizuguchi et al., 2004*; *Luk et al., 2010*). SWR1 recruitment to nucleosome-deficient or nucleosome-free regions (NFRs) of yeast promoters is due to its preference for nucleosomes adjoining long linker DNA (*Ranjan et al., 2013*), but post-recruitment activation of the SWR1 complex requires binding of both its natural substrates—the canonical nucleosome and the H2A.Z-H2B dimer—which also serve as essential activators of SWR1 (*Luk et al., 2010*). Progression of the SWR1-mediated reaction on the canonical 'AA' nucleosome generates the 'AZ' and 'ZZ' nucleosome states consecutively, which leads to repression of the ATPase and histone exchange activities of SWR1 by the H2A.Z-nucleosome end-product, thereby preventing futile expenditure of chemical energy (*Luk et al., 2010*). Specific α-C helix residues of H2A.Z on the free H2A.Z-H2B dimer are critical for its SWR1-activating function (*Clarkson et al., 1999*; *Wu et al., 2005*). However, the key-activating elements of the canonical nucleosome that distinguish it from the non-activating H2A.Z-nucleosome have been obscure. Here, we show that the histone-fold, but not the α-C helix of histone H2A in the nucleosome, has a major role in the activation of the SWR1 complex. We also define a local DNA site on the nucleosome core particle that is critical for activating SWR1.

**eLife digest** A DNA molecule can be several meters long and to fit this length inside a cell, it is wrapped around proteins called histones. This compacts the DNA to form a structure known as chromatin; complexes of DNA and histones, called nucleosomes, serve as the building blocks of chromatin.

Cells regulate the organization of chromatin to switch genes 'on' and 'off'. Complexes of proteins, such as SWR1, alter the packing of chromatin and are known as 'chromatin modifiers'. To express a gene, parts of the chromatin have to unpack to allow various proteins and other factors to access to the underlying DNA. Chromatin remodeling enzymes can loosen chromatin by sliding nucleosomes away from each other, removing them altogether, or replacing one type of histone with another. For example, a histone variant called H2A.Z appears to poise genes for expression and is enriched near the start sites of most genes in the genome. The SWR1 complex evicts the conventional, 'canonical histone' called H2A that is already present at these sites and replaces them with H2A.Z.

H2A.Z is related to H2A, and the SWR1 complex can interact with both of these proteins. However, it remains poorly understood how SWR1 can discriminate between the two at the molecular level. Ranjan et al. have now addressed this in budding yeast cells, by constructing hybrids that contain parts of H2A combined with H2A.Z. The experiments revealed that the SWR1 complex recognizes key elements within the histone H2A protein itself that differ from H2A.Z. Binding to H2A activates SWR1 and causes it to replace H2A with H2A.Z.

Ranjan et al. next looked to see if the SWR1 complex also interacts with the DNA present within a nucleosome and whether any gaps in the DNA interfere with histone replacement. The experiments revealed that gaps in DNA at a specific region of the nucleosome prevent SWR1 from depositing H2A.Z. Therefore, close contact between SWR1 and a nucleosome's DNA is another factor that is required for SWR1 activity. These findings provide new insights as to how SWR1 recognizes histone and DNA elements of a canonical nucleosome. Further work is needed to understand how SWR1 acts to replace H2A with H2A.Z.

## Results

To identify elements of histone H2A on the canonical nucleosome that activate SWR1, we constructed hybrids in which H2A segments were systematically interchanged with segments of histone H2A.Z. Hybrid nucleosome substrates were reconstituted and analyzed by a SWR1-mediated histone H2A.Z replacement assay. Interchanging the M6 and α-C regions of H2A with H2A.Z on the nucleosome had only a small effect on SWR1 activity as measured by nucleosomal incorporation of H2A.Z-3xFlag, which generates a native gel mobility upshift (*Figure 1A*). Strikingly, an additional interchange of the M4 domain of nucleosomal H2A with H2A.Z caused a ~90% decrease in H2A.Z replacement by SWR1, and extension of the interchange to M2 and M3 domains further reduced SWR1's activity (*Figure 1A*). Interchanging the M4 domain alone caused a large reduction in activity of SWR1, and activity was also reduced by interchange of M5 and M3A domains individually, whereas M2 and M3B domain interchanges had minimal effects (*Figure 1B,C*). Thus, residues contained entirely within the H2A histone-fold motif (the α1 helix, α2 helix, and loop 2) contribute to activation of SWR1. Underlying mechanism(s) could include improved enzyme binding, as the non-activating H2A.Z-nucleosome shows slightly decreased affinity for SWR1 (*Figure 1—figure supplement 1*), but activation at a post-recruitment step is required, because neither ATPase stimulation nor histone replacement occurs under saturating conditions ([*Luk et al., 2010*] and data not shown). Interchange of the M4 region of H2A for H2A.Z (substitution of five residues) in yeast causes lethality (*Figure 1—figure supplement 2A*). This indicates that the N-terminal portion of the H2A α2 helix provides an essential function apart from regulating SWR1 activity, which itself is not essential for viability. Glycine 47 is the only surface accessible H2A-specific residue in the M4 region. Strains bearing single or double amino acid substitutions to corresponding H2A.Z residues-G47K and P49A are viable. The single P49A substitution showed no reduction (even an increase) of H2A.Z levels at gene promoters by ChIP-PCR. However, the G47K interchange in the M4 region shows reduced H2A.Z incorporation (average 63% of WT), as does the double-substitution G47K, P49A (average 54% of WT), suggesting that G47 facilitates activation of SWR1 in vivo (*Figure 1—figure supplement 2B*).

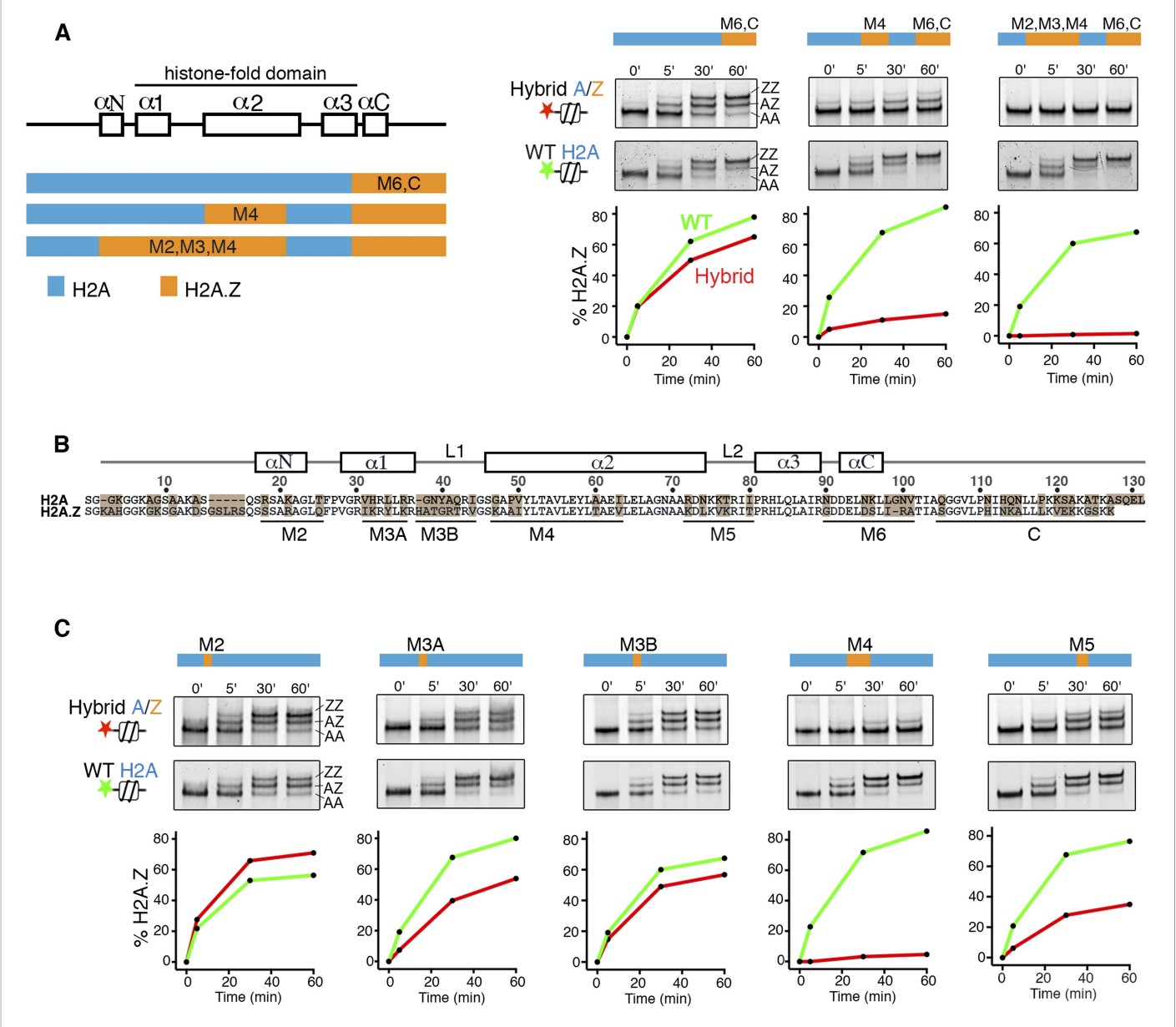

**Figure 1**. H2A histone regions in the canonical nucleosome that activate H2A.Z replacement by SWR1. (**A**) *Left*: H2A/H2A.Z hybrid histones used for reconstituting nucleosomes. *Right*: Histone H2A.Z replacement assay. Hybrid (red) and WT (green) nucleosomes (2.5 nM) were incubated with SWR1 (2 nM), H2A.Z-3F-H2B (22 nM), and ATP (1 mM) for the indicated times, and nucleosomes containing zero (AA), one (AZ), or two copies (ZZ) of H2A.Z-3F were resolved by 6% native PAGE. Top: EMSA (electrophoretic mobility shift assay) and fluorescence imaging. Bottom: H2A.Z incorporation curves. (**B**) Sequence alignment of histone H2A and H2A.Z from budding yeast. (**C**) Histone replacement assay as above, with hybrid nucleosomes containing fine H2A/H2A.Z interchanges. Bottom: H2A.Z incorporation curves.

The following figure supplements are available for figure 1:

**Figure supplement 1**. SWR1 binding to nucleosome core particles containing H2A or H2A.Z histone.

**Figure supplement 2**. Effect of H2A M4 on H2A.Z enrichment at gene promoters.

**Figure supplement 3**. Nucleosome structure showing critical H2A residues that effect SWR1 activity.

We next investigated the role of nucleosomal DNA. Previous studies have established the importance of specific DNA contacts by ATP-dependent chromatin remodelers (*Mueller-Planitz et al., 2013*; *Bartholomew, 2014*). In vitro, SWR1 is known to bind preferentially to long linker DNA adjacent to a nucleosome core particle (*Ranjan et al., 2013*). To favor SWR1 binding in one orientation, we reconstituted mono-nucleosomes bearing only one 60 bp linker and subjected bound nucleosomes to hydroxyl radical footprinting (*Figure 2A*). Notably, we observed protection at

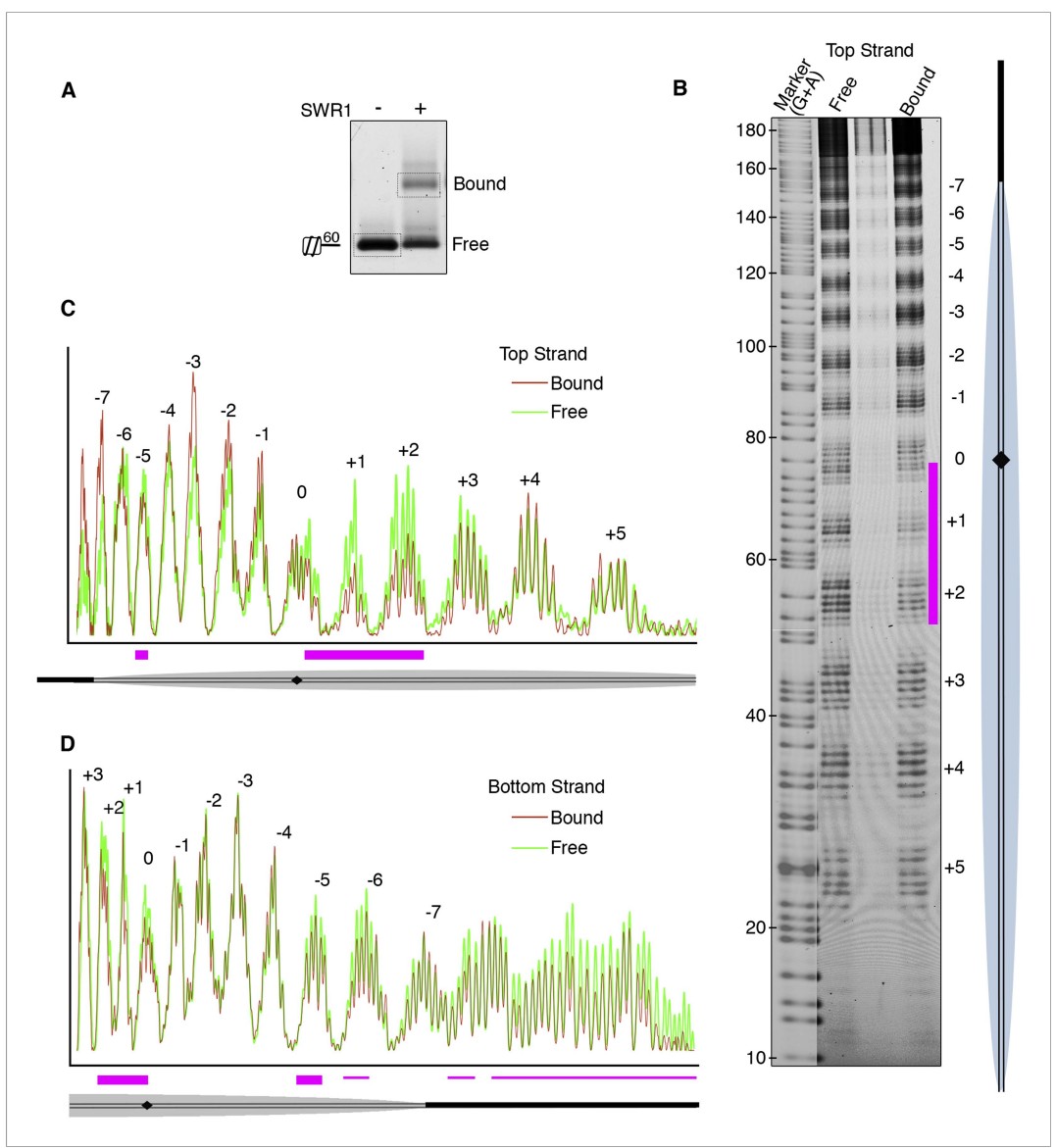

**Figure 2**. Hydroxyl radical footprinting of SWR1 on nucleosomal DNA. (**A**) EMSA (1.3% agarose gel) shows SWR1 (12 pmole; 240 nM) binding to a fluorescent end-labeled asymmetric 60 bp long linker nucleosome (7.4 pmole; 150 nM) after reaction with hydroxyl radical. (**B**) DNA samples resolved on 8% sequencing gel. Top strand is fluorescently labeled. The strongest protected area is shown as magenta bar. (**C**) Top strand intensity plots of free (green) and bound (red) nucleosome corresponding to **B**. (**D**) Bottom strand intensity plots for free and bound samples were normalized to signals at +2 and +3 SHL from dyad axis.

The following figure supplement is available for figure 2:

**Figure supplement 1**. Position of SWR1 footprint on linker-distal face of nucleosome.

superhelix locations (SHLs) SHL0, SHL+1, and SHL+2 on the linker-distal side of the nucleosome dyad (*Figure 2B,C*; *Figure 2—figure supplement 1*). Strongest protection was observed at SHL2, where other ATP-dependent chromatin remodelers have been shown to interact with the nucleosome, but on the linker-proximal or both sides of dyad (*Bartholomew, 2014*). We also observed broad protection from hydroxyl radical cleavage of long linker DNA by SWR1 (*Figure 2D*), consistent with previous findings (*Ranjan et al., 2013*).

Furthermore, we examined how gaps in nucleosomal DNA interfere with histone replacement by SWR1. Systematic introduction of two-nucleotide gaps on one DNA strand of a nucleosome showed that a single gap at −17, −18 nt from the nucleosome dyad blocked the second round of H2A.Z replacement (*Figure 3A*). Further scanning identified a 6 nt region (−17 to −22 nt from the dyad), whose integrity is required for histone replacement (*Figure 3B*). This gap-sensitive region overlaps with the hydroxyl radical footprint of linker-oriented SWR1 at nucleosome position SHL2. Introduction of gaps on both sides of the nucleosome dyad caused a complete failure of H2A.Z replacement by SWR1 (*Figure 3C*). Taken together, our findings indicate that close contact between SWR1 and nucleosomal DNA around SHL2 is critical for enzyme activation. This activation likely occurs post-recruitment, as SWR1 binding is not adversely affected on the gap-containing nucleosome substrate (*Figure 3—figure supplement 1*).

To date, all chromatin remodelers examined are able to mobilize positioned nucleosome in vitro, including the strongly positioned 601 nucleosome (*Lowary and Widom, 1998*; *Becker and Workman, 2013*; *Bartholomew, 2014*). SWR1 quantitatively evicts both H2A-H2B dimers on this nucleosome, replacing them with H2A.Z-H2B, but whether nucleosome positioning was also altered was unknown. To examine this question, we used a 601 nucleosome bearing a 60 bp linker on one side, and a native PAGE mobility assay, which separates nucleosomes on the basis of different linker lengths and spatial orientation (*Hamiche et al., 1999*). We found no substantial mobility shift indicative of a repositioned nucleosome after incorporation of (untagged) H2A.Z-H2B (*Figure 4A*). Similar results were obtained for a center-positioned 601 nucleosome (data not shown). By contrast, the INO80 remodeler mobilized the nucleosome from the end- to center-position, as shown by gel mobility shift (*Figure 4A*) (*Shen et al., 2000*; *Udugama et al., 2011*). For a more discerning technique, we mapped the precise position of AA, AZ, and ZZ nucleosomes after histone H2A.Z replacement by hydroxyl radical footprinting (*Figure 4B*). Strikingly, at single nucleotide resolution, there was no net change of the 601 nucleosome position after it underwent one or two rounds of histone H2A.Z replacement (*Figure 4C*).

## Discussion

We have identified elements of the canonical nucleosome that activate the SWR1 complex for histone H2A.Z replacement. A DNA site at SHL2 on the nucleosome, in a region identified for DNA translocation by chromatin remodelers RSC, SWI/SNF, ISW2, and ISW1, is also important for histone exchange by SWR1. The ATPase domains of the SNF2 and ISW2 nucleosome sliding complexes are known to interact with the nucleosome at SHL2 (*Dang and Bartholomew, 2007*; *Dechassa et al., 2012*), and we envision that the catalytic Swr1 ATPase also contacts the SHL2 site. Footprinting experiments show that other chromatin remodelers contact the nucleosome core particle at either the linker-proximal side or both sides of the dyad axis; however, SWR1 contacts the core particle on the linker-distal side of the dyad. This distinction between SWR1 and other remodelers may reflect the unique requirements of dimer eviction and deposition as opposed to nucleosome sliding. Within the nucleosome core particle, each H2A-H2B dimer is stabilized by histone–DNA interactions (at three minor groove locations SHL3.5, SHL4.5, and SHL5.5) and histone–histone interactions (the α2 and α3 helices of H2B interact with α2 and α3 helices of H4 in a four-helix bundle). For histone exchange, SWR1 must disrupt either one or both of these interactions, coordinated with H2A.Z-H2B deposition. This might be initiated by transient, confined DNA translocation by the SWR1 ATPase essentially as indicated for other remodelers (*Clapier and Cairns, 2009*; *Mueller-Planitz et al., 2013*), but without propagation as histone exchange is not accompanied by repositioning of the histone octamer on DNA. Alternatively, histone replacement could be initiated by a local, ATP-driven DNA conformational change near SHL2 that alters the path of the DNA superhelix, resulting in destabilization of H2A-H2B contacts with DNA or with the H3-H4 tetramer (*Figure 3—figure supplement 2*).

In budding yeast, the +1 nucleosome flanked on one side by a NFR would orient SWR1 to interact with the linker-distal face. We speculate that this configuration favors replacement of the H2A-H2B

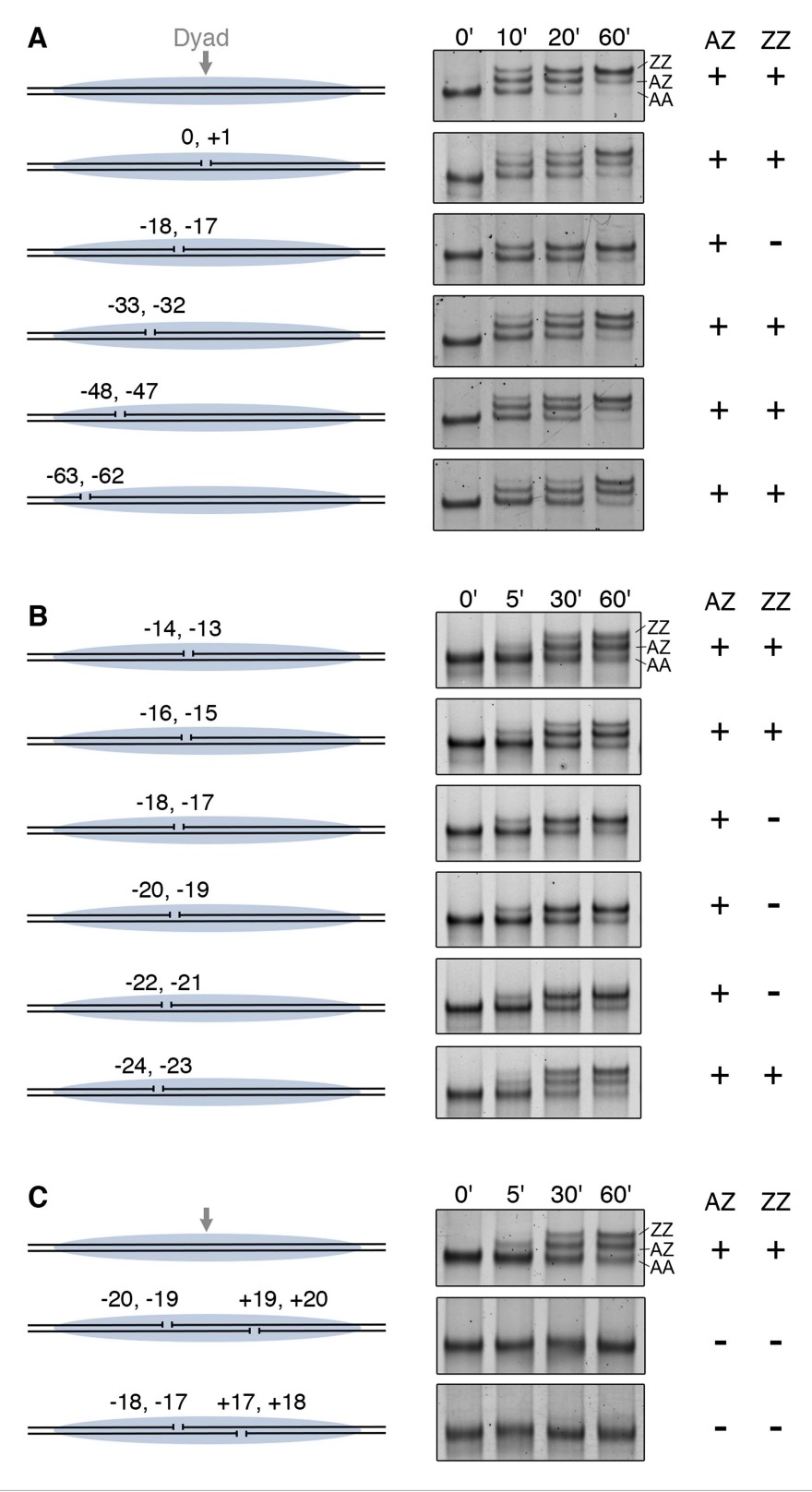

**Figure 3**. DNA gaps block SWR1 activity when positioned 17–22 bp on either side from dyad. All nucleosomes have a 20 bp linker DNA at both ends, and a two-nucleotide gap introduced at indicated positions. EMSA (6% native PAGE) shows the H2A.Z replacement reaction, terminated at the indicated times, using fluorescently labeled

*Figure 3. continued on next page*

*Figure 3. Continued*

nucleosomes (4 nM), SWR1 (2 nM), and H2A.Z-3F-H2B dimer (10 nM). Nucleosome products with 0, 1, and 2 H2A.Z-3FLAG molecules are resolved (AA, AZ, ZZ). (**A**) Mapping of gap sites that block SWR1 activity. *Left*: Design of WT and gap nucleosomes. *Right*: (+) and (−) denote presence and absence of the AZ or ZZ species. (**B**) Fine mapping of the gap-sensitive region near two turns from nucleosome dyad. (**C**) Gaps within the sensitive region on both sides of nucleosome completely block SWR1 activity.

The following figure supplements are available for figure 3:

**Figure supplement 1**. SWR1 binding to nucleosome core particle with gaps on both sides of dyad.

**Figure supplement 2**. Nucleosomal histone and DNA elements critical for SWR1 activity and model for SWR1-mediated H2A-H2B displacement.

dimer on the NFR-distal side (*Figure 2—figure supplement 1*, *Figure 3—figure supplement 2*). Consistent with this possibility, recent genome-wide sub-nucleosomal mapping shows enrichment of H2A.Z at the NFR-distal face of the +1 nucleosome (*Rhee et al., 2014*).

Earlier work has shown that the structures of H2A- and H2A.Z-containing nucleosomes show prominent differences in the region C-terminal to the histone-fold domain (*Suto et al., 2000*). This C-terminal region is important for binding of the free H2A.Z-H2B dimer to specific chaperones (*Luk et al., 2007*; *Zhou et al., 2008*; *Hong et al., 2014*), and for effector interactions post-incorporation (*Clarkson et al., 1999*; *Adam et al., 2001*). For histone H2A.Z replacement, our analysis shows that SWR1 utilizes other unique and conserved features of the H2A nucleosome for substrate specificity. Of the three SWR1-activating regions of the H2A histone-fold, the α2 helix and loop 2 are exposed on the nucleosome surface for contact with SWR1, whereas the α1 helix is buried and may act by allostery (*Figure 1—figure supplement 3A*). Residue G47 of the H2A α2 helix is highly conserved and is located at the bottom of a cleft (∼8 Å deep) on the H2A nucleosome surface (*Figure 1—figure supplement 3B*). This cleft might serve as a structural feature for recognition by SWR1; the presence of a Lysine residue at this position in H2A.Z would fill it (*Figure 1—figure supplement 3B*). It would be of interest to determine structural interactions of SWR1 with this local nucleosome surface. Our findings provide new insights on the structural basis by which canonical and H2A.Z-nucleosomes are recognized by SWR1 and should facilitate future studies of the histone H2A.Z replacement mechanism.

## Materials and methods

### Yeast strains and plasmids

Plasmid pZS66 used in this study was a gift from Zu-Wen Sun. It was made by cloning HTA1-HTB1/BamHI-SacII 2.6 kb fragment (913673–916283 sequence of chromosome IV) into the same site of pRS313 (HIS3, CEN), and pZS66 was an intermediate for pZS145 (HTA1-Flag-HTB1) (*Sun and Allis, 2002*). Yeast strain FY406 was a gift from Fred Winston and allowed mutating the sole copy of the gene-expressing histone H2A (*Hirschhorn et al., 1995*). All strains used are listed in *Supplementary file 1*.

### Nucleosome reconstitution

DNA for nucleosome preparations was PCR amplified from a plasmid containing the Widom's 601 DNA (*Lowary and Widom, 1998*). Primers labeled with Cy5, Cy3, or 6-FAM (6-carboxyfluorescein) were used for PCR. For nucleosomes with DNA gaps: primers containing deoxyuridine residues at gap sites were used for PCR amplification, and the PCR product was treated with a mix of Uracil-DNA glycosylase and endonuclease III (USER Enzyme from NEB, Ipswich, MA) (*Zofall et al., 2006*). DNA fragments with a gap have slower mobility on 6% native PAGE; and this was used to monitor completion of digestion. All DNAs, with and without gap, were PAGE-purified using a Mini Prep Cell (Bio-Rad, Hercules, CA). Recombinant core histones from yeast H2A, H2B, H2A.Z, and fly H3, H4 were purified following methods described earlier (*Luger et al., 1999*; *Vary et al., 2004*). Nucleosomes were reconstituted by salt gradient dialysis following a standard protocol (*Luger et al., 1997*).

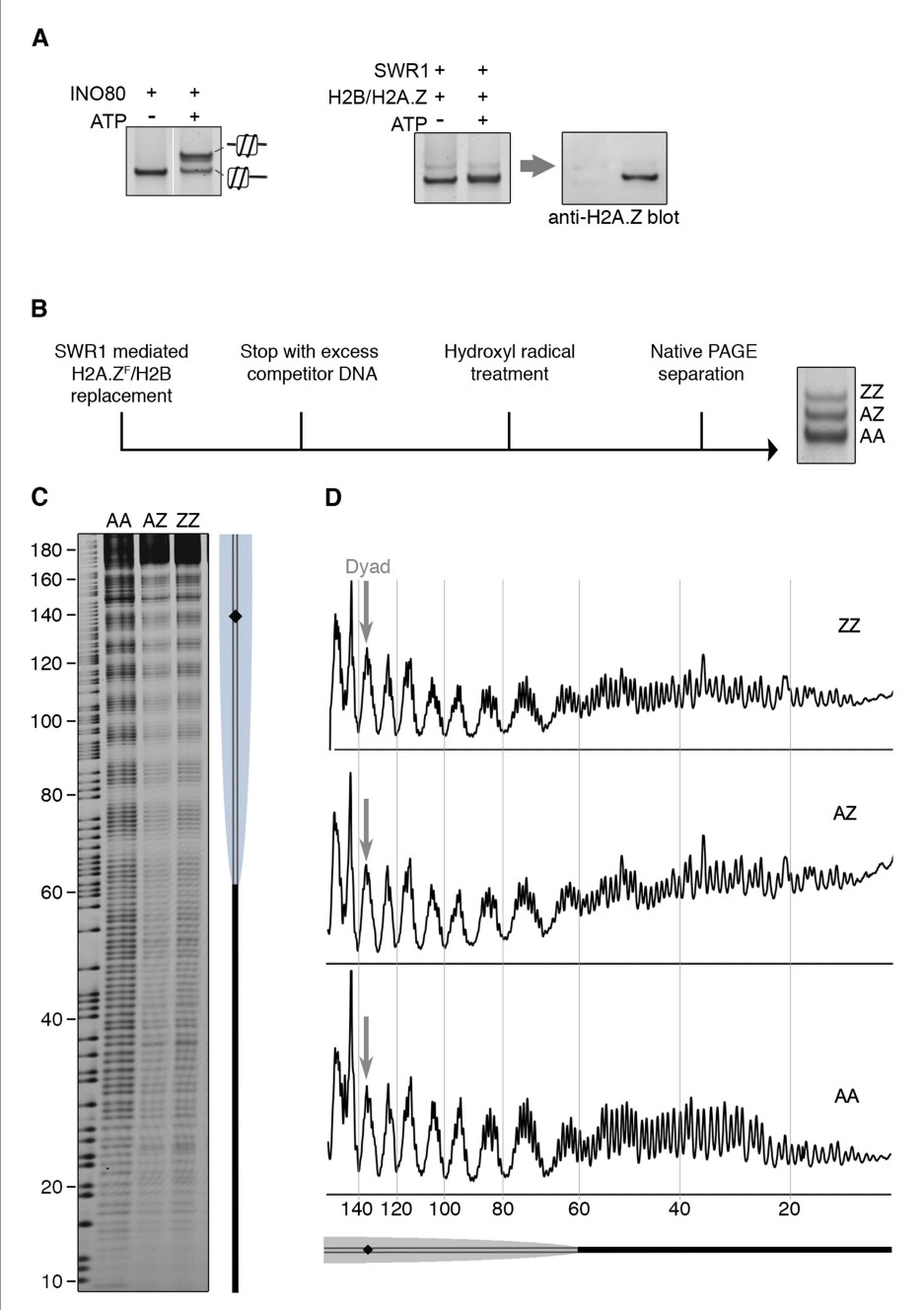

**Figure 4**. SWR1 mediates histone exchange without net change of nucleosome position. (**A**) *Left*: EMSA (6% native PAGE) shows INO80-mediated nucleosome sliding. An asymmetrically positioned 601 nucleosome with a 43 bp and 0 bp DNA linker was used for the sliding assay. *Right*: SWR1-mediated incorporation of H2A.Z-H2B dimer (without 3FLAG epitope tag). Incorporation of H2A.Z in nucleosome was confirmed by immunoblotting with anti-H2A.Z antibody. (**B**) Hydroxyl radical footprinting strategy. A canonical nucleosome with 60 bp and 0 bp linker DNA and fluorescence end-label (bottom strand) was used as substrate for histone replacement, followed by hydroxyl radical treatment and separation by 6% native PAGE. (**C**) Recovered DNA from gel slices containing AA, AZ, and ZZ states was analyzed on DNA sequencing gels. (**D**) Intensity plots for AA, AZ, and ZZ nucleosomes.

## SWR1 purification

The complex was purified as published (*Luk et al., 2010*; *Ranjan et al., 2013*). In brief, SWR1-3FLAG was affinity purified from 12-liter budding yeast cells and sedimented over a 20–50% glycerol gradient. Peak fractions were pooled and concentrated using Centricon filters (50 kDa cut-off), and the buffer was changed to 25 mM HEPES–KOH (pH 7.6), 1 mM EDTA, 2 mM $MgCl_2$, 10% glycerol, 0.01% NP-40, 0.1 M KCl. Aliquots of purified SWR1 were flash frozen and stored at −80˚C.

## SWR1 footprint on nucleosomal DNA

Hydroxyl radical footprinting was performed with minor modifications according to (*Schwanbeck et al., 2004*). Before setting up the reaction, an aliquot of purified SWR1 was thawed and buffer changed to mEX (5 mM HEPES–KOH pH 7.6, 0.3 mM EDTA, 0.3 mM EGTA, 0.01% NP40, 56 mM KCl, 5.6 mM $MgCl_2$). In a 1.5 ml tube, 12 pmole of SWR1 was mixed with 7.4 pmole 6-FAM labeled nucleosomes. Nucleosomes were in TE/50 buffer (10 mM Tris pH 7.5, 1 mM EDTA, 50 mM NaCl, and 0.4 mg/ml BSA). Typically, 20 µl nucleosome and 20 µl SWR1 were mixed and volume made up to 50 µl with mEX. On the inner wall of the tube, at different spots, the following solutions were placed: (i) 0.5 µl of 20 mM $(NH_4)FE(II)SO_4$, 40 mM EDTA, (ii) 2.5 µl of 100 mM sodium ascorbate, (iii) 0.5 µl of 3% vol/vol hydrogen peroxide. Ammonium iron (II) sulfate powder (light sensitive) was freshly dissolved in water to make 20 mM $(NH_4)FE(II)SO_4$, 40 mM EDTA solution. The sodium ascorbate solution is light sensitive and it can be stored for few weeks at 4˚C. Stock hydrogen peroxide purchased from Sigma is 30% vol/vol and it is diluted in water before use. Hydroxyl radical cleavage was initiated by spinning down reagents in microfuge and after 1 min at RT, the reaction was stopped by adding 5 µl of 100 mM thiourea, 0.5 µl of 500 mM EDTA and 8 µl sucrose loading buffer (50% sucrose in TE/50). SWR1-bound and free nucleosomes were resolved on 1.3% agarose gel in 0.2× TB. DNA from free and SWR1-bound nucleosomes was excised from gel (as shown in *Figure 2A*), and resolved on an 8% sequencing gel. FAM fluorescence signal from end-labeled DNA was scanned through sequencing glass plates on Typhoon scanner.

## H2A.Z replacement assay

The assay for SWR1-mediated H2A.Z-3FLAG/H2B incorporation in mono-nucleosomes is published (*Mizuguchi et al., 2004*; *Ranjan et al., 2013*). In brief, purified SWR1, reconstituted nucleosomes, and recombinant H2A.Z-3FLAG/H2B dimer were mixed with ATP at RT. Reactions are terminated by adding excess lambda DNA at indicated times. Incorporation of H2A.Z-3FLAG/H2B slowed the mobility of nucleosomes on 6% native PAGE, and nucleosomes containing 0, 1, or 2 copies of H2A.Z-3FLAG/H2B are resolved.

## ChIP for histone H2A.Z

ChIP follows (*Venters and Pugh, 2009*). Briefly, yeast cells with wild-type H2A or mutant H2A were grown in CSM-His medium at 30˚C to A600 = 0.7 and fixed with 1% formaldehyde at room temperature for 15 min. Chromatin was sheared by sonication and H2A.Z-HA bound chromatin was immunoprecipitated using anti-HA antibody (Clone HA-7, Sigma) and Magna ChIP Protein G beads (Millipore). Purified DNA was analyzed for enrichment of gene promoter sequences over a control sub-telomeric region on Chromosome 6 (*Kurdistani and Grunstein, 2003*) by multiplex PCR.

## Mapping nucleosome positions after H2A.Z replacement

The H2A.Z replacement reaction was set up with 100 nM 6-FAM labeled nucleosome, 40 nM SWR1-3FLAG, 400 nM H2A.Z-3XFLAG/H2B dimer, and 1 mM ATP in 50 µl mEX buffer. After 1 hr at RT, reaction was stopped by adding 4 µg of competitor salmon sperm DNA. Hydroxyl radical cleavage was performed as described above, and nucleosomes were resolved in 6% native PAGE. DNA from nucleosomes containing 0, 1, and 2 copies of H2A.Z-3FLAG/H2B was eluted and resolved on an 8% DNA sequencing gel (SequaGel 19:1 Acrylamide:BisAcrylamide, National Diagnostics).

## Supplemental information

Supplemental information includes six figures and a list of strains used.

## Acknowledgements

We thank Ralf Schwanbeck, Hua Xiao and Brian Mehl for guidance with hydroxyl radical footprinting experiments and data analysis, Alexey Shaytan, Anna Panchenko, and David Landsman (National Center for Biotechnology Information) for discussions and sharing their independent analysis of the exposed location of histone H2A G47 on the nucleosome surface. We also thank Wei-Hua Wu and Ed Luk for sharing reagents and members of our laboratory for helpful comments. This work was supported by a Leukemia and Lymphoma Society fellowship (to AR), by the Howard Hughes Medical Institute Janelia Research Campus (CW and AR), and by the Center for Cancer Research, National Cancer Institute.

## Additional information

### Funding

| Funder | Grant reference | Author |
|---|---|---|
| Howard Hughes Medical Institute (HHMI) | Janelia Research Campus | Anand Ranjan, Carl Wu |
| National Cancer Institute (NCI) | Intramural funding | Anand Ranjan, Debbie Wei, Carl Wu |

The funders had no role in study design, data collection and interpretation, or the decision to submit the work for publication.

### Author contributions

AR, Conception and design, Acquisition of data, Analysis and interpretation of data, Drafting or revising the article; FW, DW, YH, Provided Reagents; GM, Advised on plasmid shuffle and ChIP experiments; CW, Conception and design, Drafting or revising the article

## Additional files

### Supplementary file

• Supplementary file 1. Genotype of strains used in this study, related to *Figure 1*.

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
