## [Decision Letter]

Thank you for sending your work entitled “Nucleosome H2A histone-fold and DNA elements activate SWR1-mediated H2A.Z replacement” for consideration at *eLife*. Your article has been favorably evaluated by James Manley (Senior editor), a Reviewing editor, and two reviewers.

The following individuals responsible for the peer review of your submission have agreed to reveal their identity: Irwin Davidson (Reviewing editor) and Bradley Cairns (peer reviewer). A further reviewer remains anonymous.

The Reviewing editor and the reviewers discussed their comments before we reached this decision, and the Reviewing editor has assembled the following comments to help you prepare a revised submission.

Ranjan et al. searched and identified elements of the canonical nucleosome that activate the chromatin remodeling SWR1 complex for histone H2A.Z replacement. They constructed hybrids in which H2A segments were systematically interchanged with segments of histone H2A.Z. Hybrid nucleosome substrates were then reconstituted and analyzed by a SWR1-mediated histone H2A.Z replacement assay. The authors were able to show that the nucleosomal H2A α-2 helix, but not the α-C helix, plays a major role in the activation of the SWR1 complex. They also identified a local DNA site on the nucleosome core particle (around SHL2) that is critical for activating SWR1. Overall, the work is well designed, well executed, and well presented. The manuscript is clearly and concisely written. Notably, the extensive work involving DNA gaps near SHL2 shows that the reaction progresses from AA to AZ when a gap is located on one side, but is prevented (remains AA) when gaps reside on both sides (in the 17-22 region). These experiments are convincing and really inform the mechanism. Also, the apparent lack of nucleosome sliding provides an interesting contrast to other remodelers.

Specific comments:

A specific and substantive concern raised by both peer reviewers concerns the lack of in vivo data. While the biochemistry is very clean and convincing some concerns are present in the current manuscript. The most important issue is the lack of in vivo data validating the importance of the M4 box for H2A eviction and H2A.Z exchange. The authors should better define the M4 epitope and introduce the H2A M4 mutant in yeast and assess how this mutant affects H2A.Z incorporation even at a small defined set of loci. This will certainly give a broader significance to their findings.

M4 has a very strong effect (reduction) on replacement, M5 is moderate, and M3a is modest, but lacks a higher level of resolution. Having shown such a strong affect in M4, and pointing out both the conservation and positioning of G47 and P49 within this region, the work begs for an examination of their individual affects, and for a test in vivo. A limited but effective version might involve a single amino acid replacement in H2A (entirely feasible in yeast) and examination of H2A.Z replacement in vivo at a set of locations by qPCR (genome-wide not required), to establish that a reduction in replacement occurs.

The Discussion is quite short, and the general reader may not understand why interaction of SWR1 with intact DNA at SHL+2 (followed by ATP hydrolysis) is needed to remove a linker-distal H2A-H2B dimer. One can explain this by the ATPase pulling/translocating/twisting DNA from SHL+2, breaking histone-DNA contacts and therefore making the H2A-H2B dimer susceptible to release/replacement. It is clear from the positioning data that the DNA does not reposition greatly, so perhaps this pull/translocation/twist is both short and confined (in part, by SWR1 binding to the exit/linker); here, the DNA may simply collapse back to its original location following replacement by H2A.Z. The authors might have a different model in mind, but a more mechanical explanation (perhaps with a model figure) might help the reader understand more fully the interesting implications of this data.

---

## [Author Response]

*A specific and substantive concern raised by both peer reviewers concerns the lack of in vivo data. While the biochemistry is very clean and convincing some concerns are present in the current manuscript. The most important issue is the lack of in vivo data validating the importance of the M4 box for H2A eviction and H2A.Z exchange. The authors should better define the M4 epitope and introduce the H2A M4 mutant in yeast and assess how this mutant affects H2A.Z incorporation even at a small defined set of loci. This will certainly give a broader significance to their findings*.

*M4 has a very strong effect (reduction) on replacement, M5 is moderate, and M3a is modest, but lacks a higher level of resolution. Having shown such a strong affect in M4, and pointing out both the conservation and positioning of G47 and P49 within this region, the work begs for an examination of their individual affects, and for a test in vivo. A limited but effective version might involve a single amino acid replacement in H2A (entirely feasible in yeast) and examination of H2A.Z replacement in vivo at a set of locations by qPCR (genome-wide not required), to establish that a reduction in replacement occurs*.

We have performed additional experiments to address the in vivo significance of the biochemical findings. First, we interchanged the M4 region of H2A for H2A.Z (substitution of 5 residues) and examined the mutant phenotype in yeast. We found that the M4 swap causes lethality in yeast cells, indicating that the N-terminal portion of the H2A α2 helix provides an essential function apart from regulating SWR1 activity, which is not essential for cell viability. The single P49A substitution is viable, and shows no reduction (even an increase) of H2A.Z levels on a sample of yeast gene promoters by ChIP-PCR. However, the G47K interchange of the only surface-accessible residue in the M4 region shows reduced H2A.Z incorporation, as does the double-substitution G47K, P49A. Thus, our in vivo findings are consistent with, and support the importance of the M4 region of H2A in activating SWR1 in vitro. We have included a new supplemental figure, Figure 1—figure supplement 2, and discussed the data in the revised text (Results, first paragraph).

*The Discussion is quite short, and the general reader may not understand why interaction of SWR1 with intact DNA at SHL+2 (followed by ATP hydrolysis) is needed to remove a linker-distal H2A-H2B dimer. One can explain this by the ATPase pulling/translocating/twisting DNA from SHL+2, breaking histone-DNA contacts and therefore making the H2A-H2B dimer susceptible to release/replacement. It is clear from the positioning data that the DNA does not reposition greatly, so perhaps this pull/translocation/twist is both short and confined (in part, by SWR1 binding to the exit/linker); here, the DNA may simply collapse back to its original location following replacement by H2A.Z. The authors might have a different model in mind, but a more mechanical explanation (perhaps with a model figure) might help the reader understand more fully the interesting implications of this data*.

We thank the reviewers for requesting further articulation of mechanism(s), which was precluded by the word limit for the short article format. The paragraph below and model figure have been introduced in the revised text, which we hope will help the reader envision how histone replacement might occur.

“Within the nucleosome core particle each H2A-H2B dimer is stabilized by histone-DNA interactions (at three minor groove locations: SHL3.5, SHL4.5 and SHL5.5) and histone-histone interactions (α2 and α3 helices of H2B interact with α2 and α3 helices of H4 in a four-helix bundle). For histone exchange, SWR1 must disrupt either one or both of these interactions, coordinated with H2A.Z-H2B deposition. This might be initiated by transient, confined DNA translocation by the SWR1 ATPase essentially as indicated for other remodelers (6, 20), but without propagation as histone exchange is not accompanied by repositioning of the histone octamer on DNA. Alternatively, histone replacement could be initiated by a local, ATP-driven DNA conformational change near SHL2 that alters the path of the DNA superhelix, resulting in destabilization of H2A-H2B contacts with DNA or with the H3-H4 tetramer (Figure 3—figure supplement 2).”